# Induced Heteroresistance in Carbapenem-Resistant *Acinetobacter baumannii* (CRAB) via Exposure to Human Pleural Fluid (HPF) and Its Impact on Cefiderocol Susceptibility

**DOI:** 10.3390/ijms241411752

**Published:** 2023-07-21

**Authors:** Vyanka Mezcord, Jenny Escalante, Brent Nishimura, German M. Traglia, Rajnikant Sharma, Quentin Vallé, Marisel R. Tuttobene, Tomás Subils, Ingrid Marin, Fernando Pasteran, Luis A. Actis, Marcelo E. Tolmasky, Robert A. Bonomo, Gauri Rao, María S. Ramirez

**Affiliations:** 1Center for Applied Biotechnology Studies, Department of Biological Science, College of Natural Sciences and Mathematics, California State University Fullerton, Fullerton, CA 92831, USA; mezcordvyanka@csu.fullerton.edu (V.M.);; 2Unidad de Genómica y Bioinformática, Departamento de Ciencias Biológicas, CENUR Litoral Norte, Universidad de la República, Salto 50000, Uruguay; 3UNC Eshelman School of Pharmacy, University of North Carolina, Chapel Hill, NC 27599, USAquentinv@email.unc.edu (Q.V.);; 4Área Biología Molecular, Facultad de Ciencias Bioquímicas y Farmacéuticas, Universidad Nacional de Rosario, Rosario 2000, Argentina; 5Instituto de Biología Molecular y Celular de Rosario (IBR, CONICET-UNR), Rosario 2000, Argentina; 6Instituto de Procesos Biotecnológicos y Químicos de Rosario (IPROBYQ, CONICET-UNR), Rosario 2000, Argentina; 7National Regional Reference Laboratory for Antimicrobial Resistance (NRL), Servicio Antimicrobianos, Instituto Nacional de Enfermedades Infecciosas, ANLIS Dr. Carlos G. Malbrán, Buenos Aires 1282, Argentina; 8Department of Microbiology, Miami University, Oxford, OH 45056, USA; 9Research Service and GRECC, Louis Stokes Cleveland Department of Veterans Affairs Medical Center, Cleveland, OH 44106, USA; 10Departments of Medicine, Pharmacology, Molecular Biology and Microbiology, Biochemistry, Proteomics and Bioinformatics, Case Western Reserve University School of Medicine, Cleveland, OH 44106, USA; 11CWRU-Cleveland VAMC Center for Antimicrobial Resistance and Epidemiology (Case VA CARES), Cleveland, OH 44106, USA

**Keywords:** *Acinetobacter baumannii*, human pleural fluid, cefiderocol, NDM-1, carbapenem-resistance, whole-genome sequencing, antibiotic susceptibility assays, static killing-assay

## Abstract

Infections caused by Carbapenem-resistant *Acinetobacter baumannii* (CRAB) isolates, such as hospital-acquired pneumonia (HAP), bacteremia, and skin and soft tissue infections, among others, are particularly challenging to treat. Cefiderocol, a chlorocatechol-substituted siderophore antibiotic, was approved by the U.S. Food and Drug Administration (FDA) in 2019 and prescribed for the treatment of CRAB infections. Despite the initial positive treatment outcomes with this antimicrobial, recent studies reported a higher-than-average all-cause mortality rate in patients treated with cefiderocol compared to the best available therapy. The cause(s) behind these outcomes remains unconfirmed. A plausible hypothesis is heteroresistance, a phenotype characterized by the survival of a small proportion of cells in a population that is seemingly isogenic. Recent results have demonstrated that the addition of human fluids to CRAB cultures leads to cefiderocol heteroresistance. Here, we describe the molecular and phenotypic analyses of CRAB heteroresistant bacterial subpopulations to better understand the nature of the less-than-expected successful outcomes after cefiderocol treatment. Isolation of heteroresistant variants of the CRAB strain AMA40 was carried out in cultures supplemented with cefiderocol and human pleural fluid (HPF). Two AMA40 variants, AMA40 IHC1 and IHC2, were resistant to cefiderocol. To identify mutations and gene expression changes associated with cefiderocol heteroresistance, we subjected these variants to whole genome sequencing and global transcriptional analysis. We then assessed the impact of these mutations on the pharmacodynamic activity of cefiderocol via susceptibility testing, EDTA and boronic acid inhibition analysis, biofilm formation, and static time-kill assays. Heteroresistant variants AMA40 IHC1 and AMA40 IHC2 have 53 chromosomal mutations, of which 40 are common to both strains. None of the mutations occurred in genes associated with high affinity iron-uptake systems or β-lactam resistance. However, transcriptional analyses demonstrated significant modifications in levels of expression of genes associated with iron-uptake systems or β-lactam resistance. The *bla*_NDM-1_ and *bla*_ADC-2,_ as well as various iron-uptake system genes, were expressed at higher levels than the parental strain. On the other hand, the *carO* and *ompA* genes’ expression was reduced. One of the mutations common to both heteroresistant strains was mapped within *ppiA*, a gene associated with iron homeostasis in other species. Static time-kill assays demonstrated that supplementing cation-adjusted Mueller–Hinton broth with human serum albumin (HAS), the main protein component of HPF, considerably reduced cefiderocol killing activity for all three strains tested. Notably, collateral resistance to amikacin was observed in both variants. We conclude that exposing CRAB to fluids with high HSA concentrations facilitates the rise of heteroresistance associated with point mutations and transcriptional upregulation of genes coding for β-lactamases and biofilm formation. The findings from this study hold significant implications for understanding the emergence of CRAB resistance mechanisms against cefiderocol treatment. This understanding is vital for the development of treatment guidelines that can effectively address the challenges posed by CRAB infections.

## 1. Introduction

*Acinetobacter baumannii* is an opportunistic Gram-negative bacillus that is primarily responsible for causing infections among critically ill patients that may be immunocompromised [1]. The two principal clinical manifestations are pneumonia, an infection that results in the inflammation of the lungs air sacs, and bacteremia, a condition where bacteria enter the bloodstream, followed by complicated urinary tract infections (cUTIs), meningitis, traumatic or post-surgical wound infections, and osteomyelitis [2]. Carbapenem-resistant *Acinetobacter baumannii* (CRAB) was recently classified as a critical priority pathogen by the World Health Organization and the Centers for Disease Control and Prevention (CDC), as infections due to this pathogen are challenging to treat given the lack of viable treatment options [3,4]. Additionally, the global emergence and spread of *A. baumannii* strains resistant to multiple classes of antibiotics highlights the need for new antimicrobial therapies [4]. Despite efforts by several research groups and pharmaceutical companies over the past decade [5,6,7], only two novel drugs, cefiderocol and sulbactam-durlobactam, have been approved by the U.S. Food and Drug Administration (FDA) against *A. baumannii* (https://www.accessdata.fda.gov/drugsatfda_docs/label/2019/209445s000lbl.pdf, https://www.accessdata.fda.gov/drugsatfda_docs/label/2023/216974Orig1s000Correctedlbl.pdf, accessed on 19 July 2023). Guidance documents from various American and European scientific societies recommend cefiderocol for treating CRAB infections. However, these recommendations are based on in vitro results and only limited clinical trials. Although positive outcomes abound, there are recent reports indicating decreased cefiderocol efficacy against multidrug resistant (MDR) CRAB [8,9,10,11], which is suggestive of an increasing cefiderocol resistance [12,13,14].

This outcome may be suggestive of a phenomenon known as heteroresistance. Hete-roresistance is a phenotype wherein a small fraction of bacteria within a bacterial community develop resistance under antibiotic pressure [15,16]. Heteroresistance can lead to consequential resistance since the resistant subpopulation expands following prolonged antibiotic exposure. Heteroresistance to cefiderocol has been observed among different carbapenem-resistant Gram-negative species [17,18]. CREDIBLE-CR (A MultiCenter, RandomizED, Open-label ClInical Study of S-649266 or Best AvailabLE Therapy for the Treatment of Severe Infections Caused by Carbapenem-Resistant Gram-negative Pathogens), a randomized, open-label, multicenter phase 3 clinical trial involving 16 countries, was conducted to evaluate the safety and efficacy of cefiderocol vs. the best available therapy for the treatment of nosocomial pneumonia, bloodstream infection, sepsis, or complicated urinary tract infection due to carbapenem-resistant Gram-negative pathogens [19]. Among the 118 adults patients in the intent-to-treat population, *A. baumannii* infected 54 patients, making it the most frequent carbapenem-resistant pathogen [19]. Most of the previous trials that included *A. baumannii* were focused on a colistin-based regimen [8]. The cefiderocol group had a higher all-cause mortality rate (19/39), especially in patients with nosocomial pneumonia, bloodstream infection, or sepsis with *Acinetobacter* spp. at baseline [19]. Results from more recent studies also reported cefiderocol heteroresistance when *A. baumannii* was cultured in the presence of human serum albumin (HSA) or human pleural fluids (HPF) [20]. These human fluid components induced modifications in the expression level of genes related to high-affinity iron uptake systems and resistance to β-lactams [21,22,23,24,25,26]. This is supported by evidence demonstrating that most of the strains that exhibited heteroresistance, harbored the gene *bla*_PER-7_ [27,28]. Choby et al. observed a correlation between the amplification of Enterobacterales and *A. baumannii* extended-spectrum β-lactamases (ESBLs) genes and consequently the heteroresistance to cefiderocol [29]. Higher resistance levels were also observed in carbapenemase New Delhi metallo (NDM) β-lactamase -producing Enterobacterales isolates; moreover, in at least one case, the increased *bla*_NDM-5_ expression was correlated with increased cefiderocol resistance [30,31].

As mentioned before, the addition of human fluids to CRAB cultures can lead to cefiderocol heteroresistance. In this work, with the aim of gaining a better understanding of the underlying reason for this phenomenon, we carried out molecular and phenotypic analyses of two selected CRAB heteroresistant bacterial subpopulations obtained after exposure to HPF. The naturally selected variants demonstrated the acquired chromosomal mutations impacting gene coding for numerous functions, including iron metabolism. These heteroresistant variants also demonstrated an increased expression of genes related to β-lactamases, high-affinity iron uptake systems, and biofilm formation when compared to the parental strain. In contrast, transcripts of genes coding for outer membrane proteins were reduced in these mutant variants.

## 2. Results

### 2.1. Comparative Whole Genome Sequence Analysis of AMA40 and the IHC1 and IHC2 Heteroresistant Derivatives

The *A. baumannii* CRAB model strain, AMA40, susceptible to cefiderocol (MIC of 0.5–1 mg/L), harbors the carbapenem resistance gene *bla*_NDM-1_ and other relevant β-lactamase coding genes such as *bla*_ADC-2_ [20,27,31]. However, cefiderocol MIC for AMA40 exposed to HPF was higher by 5 doubling dilutions (>128 mg/L, 5 two-fold MIC increases) [20]. This is indicative of the emergence of cefiderocol resistant colonies within the inhibition ellipse (intracolonies), a response that indicates the presence of heteroresistant derivatives. To better understand some of the factors responsible for this phenomenon, the AMA40 IHC1 and IHC2 isolated strains were subjected to a global genomic comparative analysis.

A total of 53 mutations were identified in the AMA40 IHC1 and IHC2 strains compared to the parental strain. Among these mutations, 39 were observed in both variants, and 16 of these occurred within intergenic regions. An in silico analysis demonstrated that none of the mutations mapped within predicted promoter or regulatory sequences. All five mutations unique to AMA40 IHC1 and four out of the nine found in AMA40 IHC2 were intergenic (Appendix A). Among the intragenic mutations common to both variants, 1, 24 and 14 were nucleotide insertions, substitutions, or deletions, respectively. The analysis of the nucleotide substitutions revealed nine synonymous and 12 non-synonymous mutations (Appendix A). Eleven mutations affected genes coding for hypothetical proteins. In contrast, the rest of the mutations occurred within genes associated with known functions, such as *aidA* (quorum-quenching), *lptA* and *lptG* (outer membrane synthesis), *cas3* (CRISPR-associated nuclease/helicase) and others (Appendix A). The gene content of AMA40, AMA40 IHC1 and AMA40 IHC2 were identical.

It is well-known that mutations in genes coding for active iron-uptake systems play an important role in cefiderocol resistance. However, the comparative genomic analysis of the AMA40 parental strain and the AMA40 IHC1 and IHC2 heteroresistant derivatives demonstrated no nucleotide changes in genes coding for high-affinity iron acquisition functions, including *piuA, fur*, *tonB1*, *tonB2*, *tonB3*, *pirA*, *entAB*, *bauA* and *bfnH*, among others. Interestingly, the same non-synonymous mutation (S157A) within the *ppiA* gene of both heteroresistant derivatives was observed, suggesting a link between the lacking or deficient PpiA function and cefiderocol resistance. Previous work reported a potential correlation between *ppiA*, which encodes a peptidyl-prolyl cis/trans isomerase (PPIase), and iron uptake regulation [32]. Although the role of PpiA in *Acinetobacter* remains to be understood, our results indicate that this protein is involved in the decreased cefiderocol susceptibility, as reported previously [33]. NDM duplication or over-expression has been associated with decreased cefiderocol susceptibility [34,35]. However, that is not the case here, as both AMA40 IHC1 and IHC2 variants and the parent strain had an identical single copy of *bla*_NDM-1_ present.

### 2.2. Comparative Transcriptional Analysis of AMA40 and the IHC1 and IHC2 Heteroresistant Derivatives

The quantitative RT-PCR (qRT-PCR) analysis demonstrated that the expression of the β-lactamase genes *bla*_NDM-1_ and *bla*_ADC-2_ was significantly increased by 3.5- and 3-fold, respectively, in both heteroresistant strains (AMA40 IHC1 and IHC2) with respect to the parental strain (Figure 1A). Conversely, the porin coding genes *carO* and *ompA* were down regulated in the AMA40 IHC1 and IHC2 variants (Figure 1A). An assessment of the expression levels of the iron uptake genes *bauA*, *pirA*, *piuA*, *bfnH*, *exbD*, and *tonb3* demonstrated that all but *bauA* were expressed at significantly elevated levels in the AMA40 IHC1 and IHC2 derivatives (Figure 1B). While BauA, PirA, PiuA and BfnH are involved in the binding and transport of ferric siderophores, TonB-ExbB-ExbD represent the energy transducing complex needed for iron acquisition. It has been reported that although *tonB1* and *tonB2* are dispensable for ferric iron uptake, the deletion of *tonB3* led to a decrease in the intracellular iron content despite siderophore overproduction. This observation indicates that TonB3 plays a crucial role in iron uptake [36], although its role in cefiderocol transport remains to confirmed.

An increased expression of two β-lactamase genes and a decrease in the ability to penetrate the outer membrane could be the key factors contributing to the increased cefiderocol resistance levels expressed by the IHC1 and IHC2 variants. While an increased expression of high-affinity iron uptake systems is expected to increase the susceptibility to cefiderocol, the increased expression of genes coding for β-lactamases, such as NDM and ADC, and the decreased expression of porin-coding genes (*carO* and *ompA*), seems to explain the possible increased intake of cefiderocol that could contribute to the increased cefiderocol resistance observed in IHC1 and IHC2 compared to the parental strain.

### 2.3. Susceptibility Assays Suggest the Contribution of Increased Gene Expression of β-Lactamases Resulting in Increased Levels of Cefiderocol Resistance

Intracolonies observed in the inhibition ellipse while determining the cefiderocol MIC of AMA40 when exposed to HPF were subcultured and stored at −80 °C for further analyses [20]. Cefiderocol MIC determinations using two different methodologies, E-strips and microdilution assays, demonstrated that both strains have a higher level of resistance than the parental strain (Appendix A and Table 1). Furthermore, the enhanced resistance phenotype is not lost after 10 daily subcultures, suggesting that it is a stable trait. Additional susceptibility testing of IHC1 and IHC2 to other antibiotics (meropenem, imipenem, gentamicin, ampicillin/sulbactam, amikacin, ciprofloxacin, levofloxacin, tigecycline, colistin, and trimethoprim-sulfamethoxazole) was performed to determine the potential cross-resistance to cefiderocol. As expected, the strains were highly resistant to most of the antibiotic families evaluated (Table 1). However, only a 2- to 3-fold increase in MICs for colistin and amikacin, respectively, was observed in both heteroresistant variants with respect to the parental strain (Table 1). The increased resistance in colistin can be attributed in part to the mutations in the genes involved in LPS transport (*lptA* and *lptG*) and the down-regulation of *ompA.* In addition, a 4-fold increase in levofloxacin MIC was observed only for AMA40 IHC2 (Table 1). The observed unique hypothetical protein mutations observed in the IHC2 derivative could be playing a role in the increase MIC to levofloxacin. In addition, MICs of cefiderocol in combination with β-lactamase inhibitors such as avibactam, relebactam, or zidebactam were reduced. These results demonstrated the ability of β-lactamase inhibitors in restoring the susceptibility to cefiderocol to levels similar to those displayed by the parental strain (Appendix A). Additionally, zinc supplementation was performed to evaluate whether the addition of this metal could result in improved cefiderocol susceptibility, as the NDM-1 activity depends on it [37]. A 3-fold increase in cefiderocol MIC was observed in the wild-type strain when CAMHA was supplemented with 300 mg/L of ZnSO_4_. However, these changes were not observed with IHC1 and IHC2, where NDM-1 was already overexpressed based on our transcriptional analysis (Appendix A).

EDTA and boronic acid assays were performed to evaluate the contribution of metallo-β-lactamases (*bla*_NDM-1_) or class C β-lactamases (*bla*_ADC-2_) towards reducing cefiderocol susceptibility. In both assays, no significant changes were observed for the AMA40 parental strain; however, a slight increase in the halo was observed for the heteroresistant strains (Appendix A).

In sum, these results demonstrated an increased resistance to colistin and amikacin in both mutant strains, while increased resistance to levofloxacin was only observed in IHC2. In addition, the role of *bla*_NDM-1_ and *bla*_ADC-2_ in the increase in cefiderocol resistance was supported by a transcriptional analysis as well as phenotypic assays.

### 2.4. Static Time-Kill Studies Demonstrated Reduced Cefiderocol Killing Activity in the Presence of HSA

Previous studies have demonstrated that the presence of HSA can lead to an increase in cefiderocol MICs [20]. Static time-kill studies were conducted in the presence and absence of a physiologically relevant concentration of 3.5% HSA [37,38]. The time to reach maximum carrying capacity is strain specific and is also based on the growth media and the presence/absence of HSA. In the absence of HSA, all three isolates reached a maximum carrying capacity by ~6 h with a bacterial density of 8.5, 8.6 and 8.9 Log CFU/mL for AMA40, AMA40 IHC1 and AMA40 IHC2, respectively (Figure 2A–C). When CAMHB was supplemented with HSA, the maximum carrying capacity was reached earlier (at 4 h) with a lower bacterial density when compared without HSA, 8.2, 8.1 and 8 for AMA40, AMA40 IHC1 and AMA40 IHC2, respectively (Figure 2D–F).

In the absence of HSA, cefiderocol concentrations >1 mg/mL against both AMA40 and AMA40 IHC2, resulted in >2 Log_10_ CFU/mL reduction in bacterial burden (Figure 2A,B and Appendix A) by 8 h. In the presence of 3.5% HSA, cefiderocol activity was considerably reduced against both strains (Appendix A). The addition of 3.5% HSA resulted in a 1.1 Log_10_ CFU/mL at 8 h with cefiderocol 8 mg/mL against AMA40 alone, and none of the cefiderocol concentrations demonstrated any effect against AMA40 IHC2. Notably, none of the cefiderocol concentrations tested were effective against IHC1; the killing activity observed with this derivative was similar to the growth controls either in the presence (Figure 2C) or absence (Figure 2F) of HSA.

### 2.5. Increased Biofilm Formation by the Cefiderocol Heteroresistant CRAB Cells

Biofilms are responsible for the persistence of bacterial infections associated with foreign bodies such as catheters or prothesis [39]. Hence, determining whether there is a change in the ability to form biofilms by heteroresistant AMA40 IHC1 and IHC2 derivatives provides information about their pathogenicity. Both strains produced a significantly greater biofilm than the parental AMA40 strain based on the quantification of biofilm production [40] (Figure 3A).

To determine whether changes at the biofilm formation phenotype level were correlated with modifications at the transcriptional level, qRT-PCR analysis of biofilm-related genes was carried out. The expression of the genes encoding for the CsuAB fimbrial major subunit and the CsuE component [41,42] as well as the BfmR response regulator was increased by 2- to 3-fold in the IHC1 and IHC2 when compared with the AMA40 wild-type strain (Figure 3B). The transcriptional expression of *csuB* and *pilT* genes was also increased in the IHC1 and IHC2 strain, but not significantly.

In sum, all these results demonstrated that the heteroresistant cells exhibit an increased biofilm formation capacity with a concomitant-increased expression of genes associated with biofilm production. The increased biofilm formation can be contributing to the increased cefiderocol resistance observed in both AMA40 IHC1 and IHC2. An additional interesting observation is the increased expression of *bfmR*. The two-component regulatory system BfmRS is known to have a role in determining a variety of *A. baumannii* responses including protection against β-lactam antibiotics [43,44,45]. This brings up the question whether BfmRS can be playing a role in cefiderocol heteroresistance.

## 3. Discussion

(a)Comparison with previous findings related to cefiderocol resistance mechanism:

Cefiderocol is demonstrated to be a promising new option for hard-to-treat infections caused by carbapenem-resistant Gram-negative bacilli, including *A. baumannii.* However, there have been increasing reports of cefiderocol resistance [8,9,13,14]. In the present study, we demonstrated the emergence of heteroresistant AMA40 CRAB cells after exposure to HSA-containing human fluids. The genomic, transcriptional, and phenotypic analysis of the two randomly selected isogenic variants indicated that multiple factors may be responsible for the cefiderocol resistance phenotype of IHC1 and IHC2 derivatives, including: (i) genomic mutations, (ii) increased expression of β-lactamases, (iii) reduced expression of porins, and (iv) increased biofilm formation. The *ppiA* mutation is an interesting observation that may be related to the increased cefiderocol resistance of the aforementioned AMA40 derivatives. In *Mycobacterium tuberculosis*, PpiA is upregulated during heat shock, implying that it may be related to stress responses and possibly virulence [46]. The *M. tuberculosis ppiA* gene was also downregulated during iron depletion, suggesting that its expression could be iron regulated [47]. In other studies, PPIases demonstrated a pivotal role in catalyzing the correct folding of many prokaryotic and eukaryotic proteins involved in diverse biological functions, ranging from cell cycle regulation to bacterial infection [48]. However, the biological role of the *A. baumannii* PpiA ortholog in iron homeostasis, virulence and cefiderocol resistance remains be explored.

It has been reported that one factor that can contribute to cefiderocol resistance is the increased expression of β-lactamases. Simner et al. [34] reported a case of a transplant recipient infected with an *E. coli* isolate harboring a *bla*_NDM-5_ gene, which progressively lost susceptibility to cefiderocol following treatment. The analysis of different isolates recovered during the course of antibiotic treatment demonstrated an increase in the copy number and expression of *bla*_NDM-5_ [34]. A previously reported case of a male patient in his 50s, whose initial blood cultures had revealed a susceptible *K. pneumoniae* that became resistant to cefiderocol upon completing cefiderocol therapy, provides further evidence about the role of this gene in cefiderocol resistance. The sequencing of this *K. pneumoniae* isolate identified *bla*_NDM-5_, suggesting that the presence of NDM can be implicated in the development of cefiderocol resistance [15]. In addition, Choby et al. [17] observed the amplification of the ESBL genes in Enterobacterales and *A. baumannii* and the consequent development of heteroresistance to cefiderocol. Altogether, the previous reports and the increased expression of β-lactamases support our observation, with the AMA40 IHC1 and IHC2 strains identifying *bla*_NDM-1_ and *bla*_ADC_ as potential contributors to heteroresistance development [29].

Additional factors that could play a role in the increased resistance observed in the AMA40 heteroresistant colonies include the down-regulation of the porin-coding genes *carO* and *ompA*. CarO allows the permeation of imipenem in *A. baumannii* [49], while the lack of a functional OmpA is associated with increased susceptibility to different antibiotics such as chloramphenicol, colistin, aztreonam, imipenem, gentamicin and nalidixic acid in this pathogen [50]. Another factor that needs to be considered is the increased expression of biofilm-associated genes, with the concomitant increase in biofilm formation in both heteroresistant strains. We also observed an increase in the expression of *bfmR*. There is significant published literature describing the role of the BfmRS two-component system controlling various *A. baumannii* cellular processes, including biofilm formation [42,51]. Previous studies have also demonstrated that hyperactive alleles of BfmRS conferred increased resistance and tolerance against an expansive set of antibiotics, including dramatic protection from β-lactam activity [43,45,51,52]. The increased expression of *bfmR* observed in the heteroresistant cells could be responsible for the increase in colistin and amikacin MICs, as reported [51]. Given its role in developing heteroresistance to cefiderocol, further mechanistic studies characterizing the role BfmRS plays in cefiderocol resistance are necessary. Due to the multifactorial nature of the cefiderocol resistance observed in the strains in this study, which agree with the findings in the literature, it is impossible to assign a specific contribution of each individual mechanism.

Recently, unstable *A. baumannii* heteroresistant subpopulations were found in 8/10 samples cultured in the presence of high cefiderocol concentrations. Genomic analyses of heteroresistant isolates revealed mutations in the genes coding for PBP3 and TonB3 that were shared by all strains regardless of their resistance phenotype [18]. In contrast, the heteroresistance traits of the AMA40 IHC1 and IHC2 derivatives isolated during our work, which represent subpopulations obtained after the exposure of AMA40 to HSA-containing fluids, were maintained in a stable manner, even in the absence of cefiderocol selection pressure. This event demonstrated the plasticity of *A. baumannii* to adapt to antibiotic pressure to overcome cefiderocol treatment. Furthermore, the genomic analysis of the AMA40 IHC1 and IHC2 derivatives did not reveal a direct and clear connection to the functional expression of high-affinity iron acquisition processes. Taken together, these observations suggest that a combination of different cellular mechanisms are involved in driving the emergence of stable cefiderocol heteroresistance in a process that is affected by the presence of host fluids containing HSA.

Fortunately, several authors reported that the combination of cefiderocol and a diazabicyclooctane (DBO) derivative, such as avibactam, relebactam or zidebactam, seems to restore the antibacterial activity of cefiderocol against CRAB, at concentrations that are several times lower than its cefiderocol MIC and limits, in some cases, the emergence of resistance [18,53]. Subpopulations with a moderate to high level of resistance to cefiderocol described in this work, recovered a susceptibility to cefiderocol regardless of its combination with DBO. The mechanism of this synergistic activity of cefiderocol in combination with DBO is not understood, especially given the possibility that multiple factors are responsible for the emergence of cefiderocol-resistant subpopulations. In our work, we observed that, even in the case where the hyperproduction of β-lactamases such as *bla*_NDM_ and *bla*_ADC_ that are not inhibited or are unresponsive to DBOs, the susceptibility to cefiderocol is restored in combination with DBOs. These results further support the concept that combinatorial therapy is a good option to restore cefiderocol susceptibility while preventing the emergence of heteroresistance or resistant intra-colonies.

(b)Implication of the new findings:

The antimicrobial failure and the development of resistance by CRAB and other microbial pathogens was raised during studies that evaluated the efficacy of cefiderocol activity [19,54]. Falcone et al. observed that among patients who experienced treatment failure following cefiderocol monotherapy treatment, all had bloodstream infections (30% of Blood Stream Infection patients) [54]. In the presence of HSA, the main serum protein, the killing activity of cefiderocol was significantly reduced against both susceptible and low-level resistant strains, as observed in an in vitro model [20]. Although a reduction in the free fraction of cefiderocol available is expected due to its strong binding to HSA (*ca* 60%) [55], the antibiotic concentrations tested by far exceeded the MIC of the parental strain. In a real scenario, a significant benefit of cefiderocol treatment in patients with CRAB infections was noticed, except in VAP patients [54]. We previously demonstrated that HSA and HPF modulate the expression of genes associated with iron uptake systems and antibiotic resistance [20,22,25,56,57]. Due to its potent activity against challenging microorganisms, cefiderocol should be reserved for cases where alternative treatment options are limited. This precaution is especially important given the findings of this study, which highlight the potential development of resistance in infections such as bloodstream and pneumonia infections, where HSA is the predominant protein. The emergence of resistance in these contexts poses a latent threat and underscores the need for the cautious and judicious use of cefiderocol to preserve its effectiveness as a valuable treatment option.

(c)Future research directions:

To design more effective strategies for treating *A. baumannii* infections, future studies should be directed towards investigating the mechanisms by which HSA-rich human fluids induce cefiderocol resistance. Additionally, exploring the role of *ppiA* in cefiderocol resistance would contribute valuable insights. The use of in vivo studies would provide a better understanding of the host and bacterial factors involved in the emergence of cefiderocol resistance and guide the development of improved therapeutic approaches against *A. baumannii* infections.

## 4. Materials and Methods

### 4.1. Bacterial Strains

The carbapenem-resistant clinical *A. baumannii* AMA40 (*bla*_NDM-1_, *bla*_ADC-2,_ and *bla*_OXA-51_) strain [27,58] was used in this study. The AMA40 IHC1 and IHC2 cefiderocol heteroresistant strains, which grew within the inhibition ellipse zones after exposure of the *A. baumannii* AMA40 parental strain to HPF (Appendix A), were included in the analysis. The naturally occurred AMA40 variants selected exhibited different levels of cefiderocol resistance, including low and high levels of resistance. Copies of the IHC1 and IHC2 isolates were kept at −80 °C as Luria Bertani (LB) broth containing 20% glycerol stocks that were plated on Cystine–Lactose–Electrolyte-Deficient (CLED) medium (Beckton Dickinson, Franklin Lakes, NJ, USA) and used within 24 h after overnight (16–18 h) incubation at 37 °C. The resistance phenotype stability was determined after 10 daily subcultures in CLED antibiotic-free plates.

### 4.2. Whole Genome Sequencing and Genomic Analysis

Genomic DNA was extracted using the Wizard Promega kit (Promega, Madison, WI, USA) according to manufacturer instructions. The whole genome sequencing was outsourced to a SEQCENTER sequencing service (Pittsburgh, PA, USA) and performed using NextSeq 550 Illumina technology (San Diego, CA, USA). The sequence quality was checked by FASTQC software analysis (https://www.bioinformatics.babraham.ac.uk/projects/fastqc/ accessed on 10 May 2023), followed by trimming and filtering with Trimmomatic software (version:0.40, ILLUMNACLIP:TrueSeq3-PE.fa.2:30:10; LEADING:3; TRAILING:3; SLIDINGWINDOW: 4:15; MINLEN:36) [59]. De novo sequence assembly was performed with SPAdes (version: 3.15.4, default parameters) [60] followed by a quality assessment performed with QUAST (version: 5.2.0) [61]. Genome annotation was conducted using PROKKA [62]. Variant calling was performed using the *breseq* and *gdtools* software packages (version: 0.38.1, consensus mode, default parameter) [63]. Recombination regions were identified and removed by Gubbins software (version: 3.3.0, default parameters) [64]. Genes coding for high-affinity iron-uptake systems were identified using the sequences reported by Antunes et al. [65]. The analyses of gain and/or loss of genes were performed using Roary (software version 3.11.2) [66]. The raw data of genomic sequencing for the AMA40 wild type strain and the IHC1 and IHC2 derivatives have been deposited in the ArrayExpress database (accession number: E-MTAB-12444, https://www.ebi.ac.uk/ena/browser/view/PRJEB58109 accessed on 10 May 2023).

### 4.3. Transcriptional Analysis Using Quantitative RT-PCR

Overnight cultures of AMA40, IHC1 and IHC2 were diluted 1:10 in iron-depleted cation-adjusted Mueller Hinton broth (CAMHB) and incubated with agitation at 200 rpm for 18 h at 37 °C. RNA was extracted from each sample using the Direct-zol RNA Kit (Zymo Research, Irvine, CA, USA) following the manufacturer’s instructions. Total RNA extractions were performed using three independent biological replicates for each of the three tested strains. The extracted DNase-treated RNA was used to synthesize cDNA using the manufacturer’s protocol provided with the iScriptTM Reverse Transcription Supermix for qPCR reagents (Bio-Rad, Hercules, CA, USA). The cDNA concentrations were adjusted to 50 ng/μL as determined by OD_260_ using a NanoDrop spectrophotometer (Thermo Scientific, Waltham, MA, USA), and 2 μL were used to conduct qPCR using the qPCRBIO SyGreen Blue Mix Lo-ROX following the manufacturer’s protocol (PCR Biosystems, Wayne, PA, USA). The transcriptional analysis of *bla*_ADC,_
*bla*_NDM-1_, *ompA, carO, pirA, piuA, bauA, bfnH, exbD, tonB3, csuAB, csuB, csuE, bfmR* and *pilT* was conducted using specific primers (Appendix A). At least three independent cDNA replicates were tested in triplicate using the CFX96 TouchTM Real-Time PCR Detection System (Bio-Rad, Hercules, CA, USA). Data are presented as NRQ (normalized relative quantities) calculated using the qBASE method [67], with *recA* and *rpoB* genes as normalizers [68]. Differences were determined by ANOVA followed by Tukey’s multiple comparison test (*p* < 0.05) using GraphPad Prism (GraphPad software version 10.0.0, San Diego, CA, USA).

### 4.4. Susceptibility Assays

Antibiotic susceptibility assays were performed following the Clinical and Laboratory Standards Institute (CLSI) guidelines as described in the Thirty Edition informational supplement [69]. After adjustment to a 0.5 McFarland standard value, *A. baumannii* AMA40, IHC1 and IHC2 cells grown in iron-depleted CAMHA were used to perform susceptibility assays. Susceptibility assays were conducted using commercial E-strips (Liofilchem S.r.l., Roseto degli Abruzzi, Italy), as follows: strips containing 0.016–256 µg/mL of amikacin (AK), cefiderocol (CFDC), colistin (CO), cloxacillin (CX), gentamicin (GEN), and tigecycline (TGC); strips containing 0.002–32 µg/mL of ciprofloxacin (CIP), levofloxacin (LEV) and meropenem (MEM); strips containing 0.016–256 µg/mL ampicillin–sulbactman 2:1 (AMS); strips containing 0.016/4–256/4 µg/mL ceftazidime–avibactam (CZA); strips containing 0.002/4–32/4 µg/mL imipenem–relebactam (I/R); strips containing 0.016/8–256/8 µg µg/mL meropenem–vaborbactam (M/V); and strips containing 0.002–32 µg/mL trimethoprim–sulfamethoxazole 1:19 (SXT). All assays were conducted according to manufacturer instructions (https://www.liofilchem.com/images/brochure/mic_test_strip_patent/MTS51.pdf accessed on 10 January 2023).

In addition, 4 μg/mL of avibactam (Sigma-Aldrich), relevactam (Sigma-Aldrich), or zidebactam (Wockhardt) was added to CAMHA medium when indicated. Measurements were taken after plates were incubated at 37 °C for 18 h. CLSI breakpoints were used for data interpretation [69]. *Escherichia coli* ATCC 25922 was used for quality control purposes. In addition, cefiderocol MICs of AMA40, IHC1 and IHC2 were also performed using the microdilution method following CLSI guidelines. All susceptibility assays were repeated at least two times using independent biological samples each time.

### 4.5. EDTA and Boronic Acid Inhibition Assays

To determine the impact of NDM inhibition on cefiderocol susceptibility, cefiderocol disk diffusion assays with and without the addition of EDTA were performed. For this purpose, two 30-μg cefiderocol disks, one supplemented with 10 μL of 0.5 mmol/L EDTA (Sigma-Aldrich, Burlington, MA, USA), were deposited on the surface of a Mueller–Hinton agar plate inoculated with a lawn of AMA40, AMA40 IHC1 or AMA40 IHC2 cells. The cells were incubated for 18–24 h at 37 °C [70]. An increase in the growth inhibition zone > 3 mm produced by the addition of EDTA was interpreted as circumstantial evidence that NDM production was contributing to cefiderocol resistance. In addition, to evaluate *bla*_ADC_ (class C β-lactamases) contribution to cefiderocol susceptibility, CAMHA plates containing 300 μg/mL boronic acid (final concentration) were prepared following previously published recommendations [71]. Subsequently, a 30-μg cefiderocol disk was placed on the surface of a CAMHA plate inoculated with a lawn of AMA40, AMA40 IHC1, or AMA40 IHC2 cells and incubated for 18–24 h at 37 °C. An increase in the growth inhibition zone >3 mm produced by the addition of boronic acid was interpreted as circumstantial evidence that ADC production was contributing to cefiderocol resistance.

### 4.6. Static Time-Kill Studies

Static time-kill studies were performed to determine bacterial killing kinetics in the absence (growth control) and presence of cefiderocol against AMA40, AMA40 IHC1 and AMA-40 IHC2 strains. Cefiderocol killing activity was evaluated at clinically achievable concentrations (0.5, 1, 4 and 8 µg/mL) [55] with and without 3.5% HSA against an initial inoculum of 5 × 10^6^ CFU/mL. Cefiderocol was added to a log growth phase bacterial suspension. Serial samples obtained at 0, 2, 4, 6 and 8 h following the addition of the drug were diluted with normal saline, and 50 μL of the appropriate bacterial dilution were spirally plated on CAMHA using an automated spiral plater (Don Whitley WASP Touch, Microbiology International, Frederick, MD, USA) and incubated at 37 °C. Following a 24-h incubation period, bacteria were quantified using a ProtoCOL automated colony counter (Symbiosis, Cambridge, UK). The lower limit of quantification was 1.3 log_10_ CFU/mL. All assays were repeated at least three times using independent biological samples each time.

### 4.7. Biofilm Assays

Overnight cultures of AMA40, AMA40 IHC1, and AMA40 IHC2 cells grown in fresh LB medium with agitation for 18 h at 37 °C were used to determine biofilm formation. The optical density at 600 nm (OD_600_) of each culture was adjusted to 0.9–1.1 and 100 μL were placed in a 96-well polystyrene microtiter plate, which was incubated at 37 °C for 24 h without shaking. The next day, the OD_600_ was measured, using a microplate reader, to determine the total biomass. The wells were emptied, washed three times with 1× phosphate-buffered saline (PBS) and stained for 15 min with 100 μL 1% crystal violet (CV) for 15 min at room temperature. CV was dumped and plated and then washed with PBS. CV associated with biofilms attached to the plate wells was solubilized in ethanol–acetone (80:20) for 30 min at room temperature, and OD_580_ was determined for each sample. Biofilm formation was determined as the OD_580_/OD_600_ ratio to minimize growth differences among tested samples. Experiments were performed in triplicate, with at least three technical replicates per biological replicate. Wells containing sterile LB medium were used as negative controls. Statistical significance (*p* < 0.05) was determined by two-way ANOVA followed by Tukey’s multiple comparison test.

## 5. Concluding Remarks

In the present work, we found that two independent cefiderocol-heteroresistant derivatives demonstrated no mutations in genes coding for active iron acquisition or β-lactam resistance functions. However, both derivatives demonstrated the same point mutation in *ppiA*, a gene associated with iron homeostasis in other species. In addition, the *bla*_NDM-1_ and *bla*_ADC-2_ genes were expressed at higher levels in the cefiderocol heteroresistant cells that were associated with a decreased cefiderocol susceptibility. Notably, static time-kill assays demonstrated that the cefiderocol killing activity was considerably reduced in the presence of HSA. In sum, our study demonstrates that the presence of HSA-containing fluids significantly reduces *A. baumannii* susceptibility to cefiderocol. This might be due to mechanisms including genomic point mutations, phenotypic modifications such as increased biofilm formation, and changes in gene expression. Further studies focused on understanding the mechanisms through which HSA-rich human fluids elicit antibiotic resistance may provide the basis for designing more effective strategies for treating *A. baumannii* infections.

## Figures and Tables

**Figure 1 ijms-24-11752-f001:**
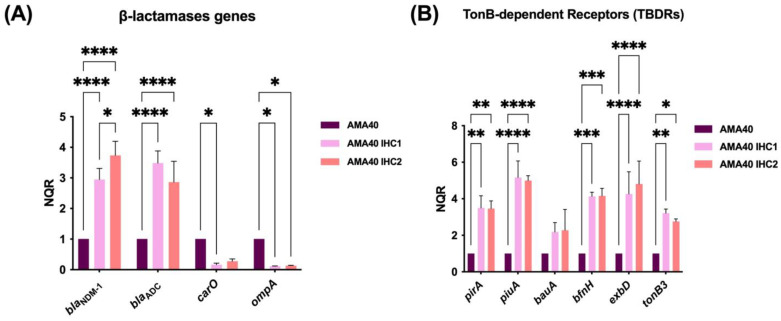
Expression of genes coding for β-lactamases, outer membrane proteins and iron uptake functions in the AMA40, AMA40 IHC1 and AMA40 IHC2 strains. qRT-PCR of *bla* genes (*bla*_ADC,_ and *bla*_NDM-1_), genes coding for the outer membrane proteins OmpA and CarO (**A**) and iron uptake-related proteins PirA, PiuA, BauA, BfnH, ExbD and TonB3 (**B**) expressed in CAMHB. The data shown are the mean ± SD of normalized relative quantities (NRQ) obtained from transcript levels. At least three independent biological samples were tested using four technical replicates for each sample. Statistical significance (*p* < 0.05) was determined by two-way ANOVA followed by Tukey’s multiple comparison test. Significance was indicated by: * *p* < 0.05, ** *p* < 0.01, *** *p* < 0.001 and **** *p* < 0.0001.

**Figure 2 ijms-24-11752-f002:**
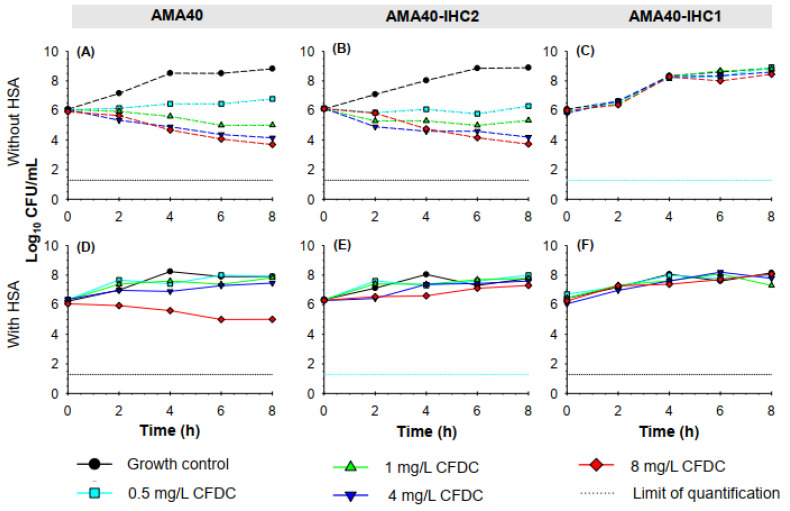
Static time-kill kinetics of cefiderocol monotherapy (0.5, 1, 4 and 8 mg/L) against an initial inoculum of 5 × 10^6^ CFU/mL of *A. baumannii* isolates AMA40 (**A**,**D**), AMA40 IHC2 (**B**,**E**) and AMA40 IHC1 (**C**,**F**) in CAMHB (**A**–**C**) or CAMHB supplemented with 3.5% HSA (**D**–**F**) over 8 h of incubation at 37 °C. The black and aqua dashed lines represent the limit of quantification for CFU/mL bacterial count (1.3 Log_10_ CFU/mL).

**Figure 3 ijms-24-11752-f003:**
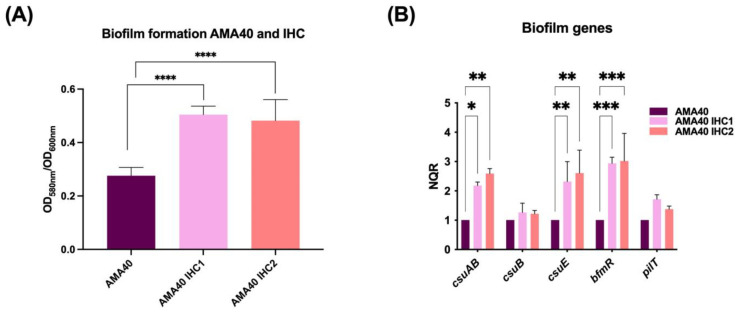
Genetic and phenotypic analysis of biofilm production. (**A**) Biofilm assays performed with *A. baumannii* AMA40, AMA40 IHC1 and AMA40 IHC2 are represented by the OD_580_/OD_600_ ratio. Statistical significance (*p* < 0.05) was determined by two-way ANOVA followed by Tukey’s multiple comparison test. Significance was indicated by: **** *p* < 0.0001. (**B**) qRT-PCR of *csuAB*, *csuB*, *csuE*, *bfmR* and *pilT* genes expressed in CAMHB. The data shown are mean ± SD of normalized relative quantities (NRQ) obtained from transcript levels. At least three independent biological samples were tested using four technical replicates. Statistical significance (*p* < 0.05) was determined by two-way ANOVA followed by Tukey’s multiple comparison test. Significance was indicated by: * *p* < 0.05, ** *p* < 0.01 and *** *p* < 0.001.

**Table 1 ijms-24-11752-t001:** Minimal Inhibitory Concentrations (MICs) of the CRAB AMA40 and heteroresistant strains, performed using MTS strips (Liofilchem S.r.l., Italy) on cation-adjusted Mueller–Hinton agar.

MICs (mg/L)														
Strain	CFDC	MEM	GEN	AK	AMS	CIP	M/V	CZA	CX	SXT	I/R	CO	TGC	LEV
AMA40	0.5	>256	>256	32	>256	>256	32	>256	>256	>32	>32	0.125	0.125	4
AMA40 IHC1	>256	>256	>256	>256	>256	>256	64	>256	>256	>32	>32	0.50	0.125	4
AMA40 IHC2	8	>256	>256	>256	>256	>256	64	>256	>256	>32	>32	0.50	0.19	>32

CFDC: cefiderocol, MEM: meropenem, GEN: gengtamicin, AK: amikacin, AMS: ampicillin–sulbactman, CIP: ciprofloxacin, M/V: meropenem–vaborbactam, CZA: ceftazidime–avibactam, CX: cloxacillin, SXT: trimethoprim–sulfamethoxazole, I/R: imipenem–relebactam, CO: colistin, TGC: tigecycline and LEV: levofloxacin.

## Data Availability

Reported and primary data could be requested to the corresponding author.

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
