# Peer review of "Induced Heteroresistance in Carbapenem-Resistant *Acinetobacter baumannii* (CRAB) via Exposure to Human Pleural Fluid (HPF) and Its Impact on Cefiderocol Susceptibility"

_ijms, 2023, doi:10.3390/ijms241411752_

Round 1
Reviewer 1 Report
In the study by Mezcord et al., Authors investigated the possible mechanisms underlying FDC heteroresistance in CRAB upon exposure to human pleural fluid (HPF). Results showed that a combination of different cellular mechanisms were possibly involved in emergence of stable cefiderocol heteroresistance, including hyperproduction of β-lactamases, non-synonymous mutation (S157A) within ppiA, downregulation of porin coding genes carO and ompA. The topic is of interest and the study was well conceived. However, the clarity of the paper could be improved by going deeper into some points.
General comments
- I would check if mutations occurring in intergenic regions (l.137.138) could affect some promoter sequences/regulators, possibly explaining the differential overexpression profiles of AMA40 IHC1 and IHC2 variants vs the parental staing, with respect to blaNDM-1/blaADC-2/carO/ompA and so on.
- although the role of blaNDM-1 and blaADC-2 in the increase cefiderocol resistance was supported by transcriptional analysis, evidence from phenotypic assays could be improved. It could be worth measuring MICs to FDC, MEM and IMI using extended (wide) ranges to see if any difference is appreciable between IHC1, IHC2 and the parental strain.
- It is not clear why IHC1 had a FDC MIC several times higher that that of IHC2, although both strains showed similar genotypic features (i.e. overexpression of NDM/ADC, iron uptake systems, reduced production of porins).
- Authors should also better explain why IHC1 and IHC2 has both higher COL MICs (and LEV for IHC2) compared to the parental strain.
Specific comments:
l.80-82: I would add sulbactam-durlobactam, as per recent FDA approval
l.137.138: I would check if mutations occurring in intergenic regions could affect some promoter sequences
l.167: possible overexpression of blaOXA-51?
l. 170.173: no mention to locus for fimsbactins biosynthesis and transport. Were they absent? A comment would be welcomed.
l.171: to which iron uptake system (if any) does the tonB3 product refer to? Please specify.
l.208: relebactam (typo)
Figure 3 may be part of supplementary material
Author Response
We appreciate the reviewer's comments. Here we send point by point our response point by point, Thanks
General comments
- I would check if mutations occurring in intergenic regions (l.137.138) could affect some promoter sequences/regulators, possibly explaining the differential overexpression profiles of AMA40 IHC1 and IHC2 variants vs the parental staing, with respect to blaNDM-1/blaADC-2/carO/ompA and so on.
Authors’ response:
As suggested, we analyzed the promoter sequences/regulators of blaNDM-1, blaADC-2, carO, ompA, as well as piuA, pirA, bauA, bfnh, and tonB3. No mutations were observed in these regions of the AMA40 IHC1 and IHC2 variants.This information was included in the revised version of the mansucript.
- although the role of blaNDM-1 and blaADC-2 in the increase cefiderocol resistance was supported by transcriptional analysis, evidence from phenotypic assays could be improved. It could be worth measuring MICs to FDC, MEM and IMI using extended (wide) ranges to see if any difference is appreciable between IHC1, IHC2 and the parental strain.
Authors’ response: We believe that doing this does not add any relevant information. We clearly observed a difference between the MIC values for the three different isogenic strains for cefiderocol MIC. Also, the IMI and MEM MIC values for the three strains is >256, a very high dose. Besides the clinical relevance, previous work showed that there are issues with the solubility of the drug at higher doses than tested, an outcome that affects reproducibility.
- It is not clear why IHC1 had a FDC MIC several times higher that that of IHC2, although both strains showed similar genotypic features (i.e. overexpression of NDM/ADC, iron uptake systems, reduced production of porins).
Authors’ response: The IHC derivatives were taken from different zones within the halo of inhibition, IHC1 was closed to >256 mg/L while IHC2 was closed to 32 mg/L. Our observations agree with the concept of heteroresistance where subpopulations of seemingly isogenic derivatives display a range of susceptibilities to a given antibiotic. Although we did not find an obvious reason to explain the differences in MIC values, we can hypothesize that subtle changes in the expression of the blaADC and/or carO can contribute to this difference, a possibility that remains to be tested.
- Authors should also better explain why IHC1 and IHC2 has both higher COL MICs (and LEV for IHC2) compared to the parental strain.
Authors’ response: The non-synonymous mutation identified in lptA and lptG (genes involved in LPS transport) can in part explain the increased resistance to colistin seen in both strains. In addition, the decrease expression of ompA can also contribute to the colistin MICs reported. Although we do not have experimental evidence that could explain the increase resistance seen in IHC2 for levofloxacin, it is possible to speculate that some the three unique mutations in hypothetical proteins present in the IHC2 are responsible for our observations. A sentence was added mentioning these observations.
Specific comments:
l.80-82: I would add sulbactam-durlobactam, as per recent FDA approval
Authors’ response: Added as requested.
l.137.138: I would check if mutations occurring in intergenic regions could affect some promoter sequences
Authors’ response: As recommended, we conducted a thorough analysis of all intergenic regions to check for mutations. We did not observe any intergenic mutations that are associated with promoter or regulatory sequences.
l.167: possible overexpression of blaOXA-51?
Authors’ response: According to experimental data, basal expression of OXA.51 was negligible and no changes in IHC1 and IHC2 were observed.
- 170.173: no mention to locus for fimsbactins biosynthesis and transport. Were they absent? A comment would be welcomed.
Authors’ response: The fimsbactins biosynthesis and transport gene locus was not mentioned because it is not present in the AMA40 genome.
l.171: to which iron uptake system (if any) does the tonB3 product refer to? Please specify.
Authors’ response:
In Gram-negative bacteria, active transport of ferrisiderophores and heme relies on the conserved TonB-ExbB-ExbD energy transducing complex. The A. baumannii genome invariably contains three tonB genes (tonB1, tonB2, and tonB3). It has been reported that tonB3 is essential for A. baumannii growth under conditions of iron limitation, while tonB1 and tonB2 seem to be dispensable for ferric iron uptake. Deletion of tonB3 led to a decrease in intracellular iron content despite siderophore overproduction, indicating the crucial involvement of TonB3 in iron uptake (reference 53 in revised manuscript). We included this information in the revised manuscript.
l.208: relebactam (typo)
Authors’ response: Fixed.
Figure 3 may be part of supplementary material.
Authors’ response: As suggested, we moved Figure 3 to supplementary materials as Figure S3.

Reviewer 2 Report
Dear Authors,
Upon reviewing your manuscript, I find your study to be highly intriguing and potentially impactful in its respective field. The subject matter you've chosen to explore is compelling and addresses a significant area of research.
However, for your study to truly resonate with the readers of the International Journal of Molecular Sciences and leave a lasting impression, it necessitates some major revisions.
The first area where I would advise adjustments is the structure of your paper. Please ensure it follows the specific format: Title, Abstract, Key Words, Introduction, Materials and Methods, Results, Discussion, and Conclusion. This logical progression is a standard in scientific literature and will make your research more accessible to your readers.
In terms of visual presentation, it's essential to adopt a consistent font as per the journal's formatting guidelines. Maintaining a uniform appearance throughout will present your work professionally and in line with the journal's standards.
Additionally, the manuscript requires considerable attention to its grammar and usage of English. Although the research is promising, the present language issues may create barriers to comprehension. I highly recommend revising the text, perhaps with the help of a professional editor or a fluent English speaker.
Lastly, attention must be paid to the referencing style. Make sure your citations and bibliography are formatted as per the specifications laid out by the International Journal of Molecular Sciences. This will lend credibility to your research and ensure it's easy for readers to trace your sources.
While the need for revisions is substantial, I want to emphasize that the essence of your study is of significant interest. I'm confident that with these improvements, your research will be a valuable contribution to our journal. I look forward to reviewing your revised manuscript.
Please see the Review Report in the attached file!

The manuscript needs moderate grammar and English language improvements.
Author Response
Thank you for your comments. We have considered the reviewer’s suggestions and provided responses to their questions/comments. The following is a point-by-point response to the reviewer’s comments. We hope that you now find the revised manuscript suitable for publication.

Round 2
Reviewer 1 Report
Authors have sufficiently addressed all raised issues.
Author Response
Thank you
Reviewer 2 Report
Dear Authors,
The paper can be published now in IJMS Journal.
Best Regards!
The English Language requires minor editing.
Author Response
thank you